# How Do Molecular Tweezers Bind to Proteins? Lessons from X-ray Crystallography

**DOI:** 10.3390/molecules29081764

**Published:** 2024-04-12

**Authors:** Arthur T. Porfetye, Patricia Stege, Rocio Rebollido-Rios, Daniel Hoffmann, Thomas Schrader, Ingrid R. Vetter

**Affiliations:** 1Department of Mechanistic Cell Biology, Max-Planck Institute of Molecular Physiology, Otto-Hahn-Straße 11, 44227 Dortmund, Germany; 2Faculty of Biology, University of Duisburg-Essen, Universitätsstrasse 5, 45141 Essen, Germany; 3Faculty of Chemistry, University of Duisburg-Essen, Universitätsstrasse 7, 45117 Essen, Germany

**Keywords:** protein crystallography, molecular tweezers, supramolecular chemistry, 14-3-3 proteins, Δ^1^-pyrroline-5-carboxyl-dehydrogenase (P5CDH/ALDH4A1)

## Abstract

To understand the biological relevance and mode of action of artificial protein ligands, crystal structures with their protein targets are essential. Here, we describe and investigate all known crystal structures that contain a so-called “molecular tweezer” or one of its derivatives with an attached natural ligand on the respective target protein. The aromatic ring system of these compounds is able to include lysine and arginine side chains, supported by one or two phosphate groups that are attached to the half-moon-shaped molecule. Due to their marked preference for basic amino acids and the fully reversible binding mode, molecular tweezers are able to counteract pathologic protein aggregation and are currently being developed as disease-modifying therapies against neurodegenerative diseases such as Alzheimer’s and Parkinson’s disease. We analyzed the corresponding crystal structures with 14-3-3 proteins in complex with mono- and diphosphate tweezers. Furthermore, we solved crystal structures of two different tweezer variants in complex with the enzyme Δ^1^-Pyrroline-5-carboxyl-dehydrogenase (P5CDH) and found that the tweezers are bound to a lysine and methionine side chain, respectively. The different binding modes and their implications for affinity and specificity are discussed, as well as the general problems in crystallizing protein complexes with artificial ligands.

## 1. Introduction

Supramolecular ligands have become important in addressing features on protein surfaces that are not accessible to conventional ligands. In contrast to the overwhelming majority of ligands that gain their binding energy from the interaction with deep pockets of their target proteins, supramolecular compounds are expected to bind to well-accessible protein surface structures and elements that are, in most cases, highly solvated. Protein–protein interactions (PPIs), constituting a large extended network of more than 56,000 specific protein contacts in humans, represent a prominent drug target and serve as key points for the regulation of biochemical processes [1]. Due to the enormous importance of these specific protein contacts, the entire network is today called the “interactome”. Biologists and pharmacologists are actively searching for small molecules that may act as inhibitors or stabilizers for PPIs, with the prospect of manipulating specific physiological protein functions and—if these are pathologic—developing new disease-modifying therapies [2,3]. Regrettably, contact areas between proteins are often large, the protein surface within is often rugged, and it involves mainly highly solvated polar residues, posing a formidable challenge for the rational design of small interfering agents [4]. Consequently, most active compounds, including the few established drugs, have been hitherto identified via the screening of large libraries [5]. However, the great progress in supramolecular chemistry (including two Nobel prizes) [6], computational modeling, as well as structural biology, now provides a platform for rational design, which has recently brought forward some remarkable new small protein ligands with a built-in selectivity for certain protein epitopes [7,8,9]. The ultimate proof for a postulated well-defined interaction between protein and ligand is a crystal structure because it reveals the exact conformation and the noncovalent interplay of attractive forces that underlie the specific complexation event.

However, the co-crystallization of a given protein with its specific surface binder poses a great challenge to crystallographers and is much more demanding than obtaining a crystal for a drug occupying a deep pocket, e.g., in the active site of an enzyme. The main obstacles are the relatively low affinities, especially of non-optimized compounds, the possibility of binding in multiple conformations (thus leading to weak electron density in the crystals), the high demand on the steric properties of the binding sites, and the limited space in the crystals due to crystal packing [10]. Crystallizing proteins together with artificial surface binders has been and still is a true art.

In the recent past, several crystal structures were published that show artificial ligands on protein surfaces: Cucurbiturils on the N-terminal FGG motif for protein dimerization [11], GCP motifs on 14-3-3 proteins [12], phosphonate-based molecular glues on 14-3-3 proteins [13], and calixarenes on various basic proteins [14]. All of these have in common the fact that a small synthetic ligand successfully competes with the water on a highly solvated protein surface and produces a stable complex, which is, in most cases, enthalpically driven. However, due to the above-mentioned inherent problems with supramolecular protein ligands, the number of solved crystal structures has remained quite low until today.

In this article, we present two novel crystal structures, analyze published structures, and rationalize the obtained complex geometries for a new class of macrocyclic protein ligands, so-called molecular tweezers (Figure 1). These were introduced by Klärner and Schrader in 2005 [15] and turned out to be potent binders to basic amino acids in peptides and proteins [16,17], with highly beneficial effects on pathologic protein aggregation [18] and viral infections [19]. Although the molecular weight of the parent compound CLR01 (diphosphate tweezer) is only 750 Dalton, the overall diameter of the polyaromatic core structure amounts to roughly 1 nm, and thus, threading onto an exposed lysine on a given protein likely produces steric clashes with neighboring protein molecules in a crystal. A small-molecule crystal structure of a tweezer that illustrates the dimensions of the molecule has been described in [16]. This large steric demand is also highly critical in crystal soaking experiments and has frequently led to the destruction of preformed protein crystals (author’s observation).

NMR experiments with unmodified tweezers (such as CLR01) binding to protein surfaces usually show a more or less transient binding of tweezer molecules to almost every surface-exposed lysine side chain (“protein camouflage”) [20,21,22]; however, for crystallographic analysis, the presence of a stable, highly occupied conformation of the ligand is required to be visible in the electron density. A sufficient occupancy can be achieved by increasing the concentration of the highly soluble tweezer compounds in the crystallization solution, but if too many lysine and/or arginine side chains are targeted, crystallization is prevented due to interference with the crystal contacts of the proteins (author’s observation). Thus, the concentration of the ligand must be a compromise between successful crystallization and sufficient occupancy, and the observed ligands in the crystals will be only a subset of what can bind in solution. This is reflected in the stoichiometries of complexes formed between proteins and amino acid ligands in solution (as evidenced by ITC and NMR titrations), which are usually much higher than just 1:1 [22].

Furthermore, the size and position of solvent channels in protein crystals are usually determined solely by the interaction and packing properties of the proteins, and although protein crystals can contain up to 80% solvent, the accessible space around the proteins is often limited. Another important factor is the composition of the crystallization solution, which must be fine-tuned to yield diffraction-quality crystals and might contain cations that compete with the lysines and arginines for tweezer binding.

In the following, we analyze all available crystal structures—including the structures presented in this article—with tweezer-containing compounds to elucidate the common elements of the binding behavior. Specifically, we investigate which residues or ions are the preferred guests inside the tweezer cavity, which differences occur in the binding geometry of modified tweezers (e.g., with one or two phosphate groups or with a covalently attached peptide), whether molecular tweezers require special structural epitopes (e.g., a certain backbone conformation or secondary structure element), and which factors contribute to the stabilization of the ligand in a well-defined conformation (like, e.g., neighboring molecules in the crystals). It should be noted that NMR spectroscopy and crystallography are two complementary methods, which together provide valuable information on the structure and dynamics of proteins complexed with artificial ligands.

## 2. Results

### 2.1. Analysis of Published Structures: 14-3-3–Tweezer Complexes

14-3-3 proteins are ubiquitous receptors for phosphorylated cargo proteins involved in the regulation of, e.g., kinases and lipases of important signal transduction pathways, e.g., Raf, PKA, or Phospholipase A_2_. Many of the seven 14-3-3 isoforms in human form dimers that are W-shaped and can bind one interacting peptide per monomer in a wide groove, usually as an extended stretch of residues. Seen from the standpoint of crystallography, this protein shape has the advantage of providing large solvent channels in the—usually tightly packed—crystals, thus allowing relatively large ligands to bind without having to compete with the crystal packing forces. Since each cargo protein uses a disordered region to dock onto the 14-3-3 adapter protein, most experiments modified a tweezer molecule with a related peptidic stretch as an anchor, which contributes almost the entire binding energy.

The first two structures in complex with a diphosphate tweezer molecule (CLR01) were determined in 2013 using the sigma isoform of 14-3-3 featuring 17 lysine residues [23]. The crystals were grown from a solution of 10 mg/mL 14-3-3 sigma and 1 mM CLR01 at 4 °C in 0.1 M Tris/HCl, 10% PEG 8000 and 500 mM MgCl_2_ and diffracted to only 3.15 Å (PDB code 5OEG), a resolution that makes it challenging to identify a bound ligand. The crystals could be improved to 2.35 Å (PDB code 5OEH) by switching to different crystallization conditions (1.36 M sodium citrate pH 6.0 and 15% (*v*/*v*) glycerol at 4 °C). Both crystal structures are very similar; they share the same space group with one molecule 14-3-3 per asymmetric unit, and both revealed lysine 214 as a guest molecule inside the tweezer cavity (Figure 2).

In the 14-3-3-CLR01 complex structure 5OEH, the tweezer buries 262 Å^2^ with the molecule providing the lysine 214 side chain and 197 Å^2^ with a symmetry-related 14-3-3 molecule, where the side chains of Gln221 and Arg224 are clamping the tweezer molecule (Figure 2, Appendix A). The CLR01 molecule is accommodated at the end of an alpha helix with one phosphate group bound to a shallow groove formed by the end of the canonical alpha-helix, which is—interestingly—slightly negatively charged, indicating that, in this case, shape complementarity seems to be more important than the charge. It is evident from the size comparison of the tweezer molecule with the protein that binding of the tweezer is sterically very challenging. The presence of the two phosphate moieties prevents binding to the center of an alpha-helix, since one of the phosphates of CLR01 would always clash with the helix backbone, explaining why CLR01 needs this specific secondary structure setting. Since Lys 214 is located close to the position of a natural peptide guest of 14-3-3, e.g., a C-Raf peptide, the tweezer could be used as a competitor, and excess CLR01 completely kicked out the C-Raf peptide guest from the 14-3-3 binding cleft (inhibitor of a PPI = protein–protein interaction) [23].

The second example of a tweezer molecule in complex with 14-3-3 (PDB codes 5M36 and 5M37, ζ isoform, 70% sequence identity with the σ isoform) highlights a different binding mode: two peptides derived from the 14-3-3 recognition motif of CDC25C (38 and 20 residues, respectively, from the region around the phosphorylated serine 216) and the diphosphate tweezer CLR01 (as separate molecules) were soaked into 14-3-3 crystals [24]. In the resulting structures, a tweezer is bound to an arginine side chain of the peptides, with the outside of the tweezer anchored in the relatively wide substrate-binding groove of the 14-3-3 protein (Figure 3). This serendipitous configuration rigidified the part of the peptide next to the CLR01, which would normally be disordered without the tweezer molecule, as seen in the corresponding apo 14-3-3 protein structure (PDB ID 5M35). The noncovalent contact between the tweezer and included arginine strengthens the natural 14-3-3-peptide interaction and constitutes a beautiful example of a “molecular glue”, which could also be demonstrated in solution via fluorescence polarization measurements [24].

Both in 5M36 and 5M37, additional tweezer molecules are found next to the previously discussed ones, which are held in place by packing against the first tweezer and the side chain of Asn183, as well as to the symmetry-related Gln150, Phe153, and Tyr178 side chains (Figure 3). The PDB coordinates do not contain atoms inside these CLR01 molecules, but the electron density clearly shows a sausage-like density inside the tweezer cavity. It could either be a molecule of the crystallization buffer (see Table 1), e.g., PEG, but the similarity to the density inside the peptide–arginine bound tweezer molecules makes it tempting to assume that this density corresponds to an arginine from another peptide-arginine, with the remainder of the peptide molecule being disordered and, therefore, invisible. In 5M36, the second monomer in the asymmetric unit shows the additional tweezer molecule in the same position as in the first monomer, whereas, surprisingly, this position is not occupied by a CLR01 molecule in 5M37. This is even more surprising since the crystallization conditions of both 5M36 and 5M37 are exactly similar, and the crystals even share the same space group. However, the b-axis is 14 Å shorter in 5M37 compared to 5M36, indicating that the crystal packing is substantially different. Indeed, the cleft between Asn183 of monomer B and its symmetry-related molecule is a tiny bit narrower compared to the cleft next to Asn183 in monomer A, and this is apparently already sufficient to prevent the binding of the tweezer. Instead, in 5M37 a CLR01 molecule shows a very well defined density next to the Lys74 of the second monomer in 5M37 (Figure 3) and packs to the side chain of the symmetry-related Gln161 and the neighboring backbone of Glu159. To date, 5OEH and 5M37 are the only X-ray structures with the diphosphate tweezer CLR01 bound to a lysine side chain of a folded protein.

In both 14-3-3 ζ structures 5M36 and 5M37, no tweezer is found next to lysine 212, which is equivalent to Lys214 in the sigma isoform of 14-3-3 (5OEH). This is again surprising since the sequence around the tweezer binding site of the 14-3-3 sigma is almost completely conserved in the ζ isoform (TLSE(D→E)SY**K**DSTLIMQLLRDN). However, the conformation of the Tyr213 side chain of the sigma isoform in 5OEH (Tyr211 in ζ 5M36/5M37) has changed, most likely due to the different crystal packing, which forces the Tyr211 side chain very close to Lys212 in the ζ isoforms, leaving no space for the CLR01 molecule. The exception is monomer A in 5M36, where Tyr211 is oriented in a different direction, but here, a symmetry-related molecule comes in too close with the alpha helix around Ala201, thus explaining the absence of the tweezer. Furthermore, the backbone next to Tyr211 seems to have multiple conformations, indicating a potential flexibility that also might impair binding. This illustrates the strong influence of the crystal packing on tweezer binding: Even the (usually weak) crystal packing forces seem to be sufficient to displace the tweezer from Lys212 in the ζ isoforms, which is consistent with the relatively low affinity of CLR01 to lysine and arginine side chains (in the range of 20–60 µM [25], even when the amino acid side chains are sterically freely accessible in solution.

To determine whether the tweezer molecule causes a change in crystal packing when co-crystallized with the protein, the space groups of the CLR01-bound crystals were compared to other 14-3-3 crystals with similar space groups and unit cells. The 14-3-3 sigma structure 6TL3 in complex with an estrogen receptor phosphopeptide [26] has the same space group as 5OEH and an almost identical unit cell and crystal packing, in spite of different crystallization conditions; therefore, the tweezer most likely did not affect the packing of the 5OEH crystal, at least not significantly. However, the environment of Lys214 in 5OEH is a bit different compared to 6TL3, and the side chain of Tyr213 moves slightly to accommodate the tweezer in 5OEH, accompanied by minor shifts of the 14-3-3 molecules. These minor differences could—in addition to the presence of the tweezer—also be due to the different crystallization conditions. The same applies to 14-3-3 ζ 7EXE [27], which has almost the same cell as 5M37, and the 14-3-3 ζ structures 6FN9/4FJ3 and 5M36 [23,28,29]. We conclude that the interactions of the protein molecules dominate the crystal lattice formation, and the tweezer binds to the spaces that are still exposed to solution in the crystal lattice.

The third example is a 14-3-3 sigma structure with an ExoS peptide covalently modified with a tweezer (PDB code 6Y7T) [30]. The linker was designed to bridge the distance between the N-terminus of the peptide and the tweezer when bound to Lys214 (PDB code 5OEH). Indeed, the peptide–tweezer hybrid bound as expected, with the peptide in the 14-3-3 groove and the tweezer in almost the same position as in 5OEH (Figure 3), which was again stabilized by almost the same interactions from a symmetry-related 14-3-3 molecule. Fusing the peptide to the tweezer resulted in an approximately 100-fold increase in affinity, from a relatively low affinity of 8 μM of CLR01 alone, and 46 μM of the ExoS-peptide alone, to 0.4 μM of the hybrid compound, highlighting a method that can be used to increase the affinity and specificity of the supramolecular compound. The conjugation of a protein–ligand with a signaling peptide may indeed be a general way to construct powerful and selective protein binders.

### 2.2. Binding of CLR01 to Sterically Related Sites of 14-3-3 Proteins—P5CDH as Model System

To further investigate the requirements for tweezer binding sites in proteins, a systematic search for epitopes similar to the tweezer binding site at Lys214 in the sigma variant of 14-3-3 proteins (PDB ID 5OEH) was performed using the software “EpitopeMatch” [http://www.epitopematch.org, accessed on 22 December 2015]. The epitope was selected from the 14-3-3 PDB structure 4HRU and included seven residues that surrounded the tweezer at around 5 Å (E210, D211, Y213, K214, D215, T217, and L218). The selected epitope was matched against the complete PDB database (113146 structures at the time). The best match revealed Lys464 of the mitochondrial aldehyde dehydrogenase Δ^1^-Pyrroline-5-carboxyl-dehydrogenase (P5CDH, PDB ID 3V9I) [31] as a potential candidate for a binding site for CLR01 (matched epitope: DDYKELQ with RMSD of 0.92 Å, Figure 4).

The solvent channels in the published crystal structure of P5CDH (PDB ID 3V9G) next to Lys464 were deemed to be sufficiently large to accommodate a tweezer molecule in at least two of the four chains in the asymmetric unit. As depicted in Figure 4, the putative CLR01 molecule on Lys464 would encounter a slightly more crowded environment compared to its counterpart in the 14-3-3 protein at Lys214. However, potential clashes may be alleviated by a minor repositioning of the tweezer molecule. Additionally, the negatively charged side chain of Glu492 might present an unfavorable interaction with the phosphates but could potentially reorient itself into the solvent. Therefore, P5CDH was chosen as a target for the systematic analysis of co-crystallization with two different tweezer molecules, the diphosphate tweezer CLR01, which was also used for the 14-3-3 structures, and a monophosphate tweezer, where one phosphate group is replaced by a sterically less-demanding hydroxyl group.

### 2.3. Structure of P5CDH with the Diphosphate Tweezer CLR01

P5CDH crystallizes as a tetramer consisting of a dimer of the physiologically relevant dimers (Figure 5A) with the N-termini of all four molecules in the asymmetric unit packing together in a large cleft between the two dimers formed by chains A and B, or C and D, respectively (Figure 5, Table 2). Surprisingly, the CLR01 molecule was found in only one copy per tetramer in the asymmetric unit, and instead of binding to a lysine or arginine side chain, as expected, the Met17 of chain A—which actually belongs to artificially introduced linker residues (see methods)—was seen as the guest inside the aromatic ring (Figure 5). The tweezer bound to monomer A is stabilized by packing against residues of monomer D. Interestingly, the regions around Met17 in monomers B, C, and D are disordered, indicating that the binding of CLR01 has a stabilizing effect on the N-terminus of monomer A, similar to the effect on the binding of a peptide to 14-3-3 described in the previous section. Compared to apo P5CDH, which crystallized in the same space group *P*6_5_ (PDB ID 3V9G), the liganded structure is very similar, as expected (RMSD of 0.79 Å over 1951 Cα atoms). The unit cell sizes differ by approximately 2 Å on all axes, indicating minimally different packing. Indeed, the dimers of the ligand-bound P5CDH are slightly rotated relative to each other by approximately 4° compared to the apo structure (PDB ID 3V9G). In the unliganded form, the asymmetric unit contains two two-fold rotational non-crystallographic symmetry axes, one through the physiological dimer and the other running between the dimer-of-dimers. This symmetrical arrangement is apparently altered by binding to the CLR01 molecule since otherwise, it would not fit into the narrower cleft seen in the apo crystals. Since the crystals were obtained via co-crystallization, most likely, the asymmetric tetramers (induced by CLR01 binding) were preferred for crystal growth as they apparently had a more favorable packing in the crystal lattice as compared to the tetramers with more than one CLR01 molecule bound.

The side chain of methionine 17 is an unexpected guest inside the tweezer molecule. To further validate this binding mode, the anomalous signal of phosphorous and sulfur was measured at a wavelength of 1.698 Å at the synchrotron. From these data, a density map can be calculated that shows the location of those atoms. Three peaks are observed at 2I/σI, at both phosphate groups of the tweezer and inside the ring system at a position corresponding to the sulfur atom of methionine 17. Although the anomalous signal is relatively weak, this strongly supports the binding mode of the tweezer encircling the Met17 side chain.

Figure 5D–F depict the binding mode of CLR01 to P5CDH in more detail. The molecule is tightly bound between the dimerization hairpin of monomer A comprising Pro189 and Pro190, and the N-terminus of monomer A provides the encircled Met17 residue. The phosphate group next to Ala33 forms H-bonds to the side chain of Arg23 of monomer A, as well as to Arg171 and the backbone amide of Ala33 of monomer D. Lysine residues 31 (monomer D) and 552 (monomer A) are clamping the outsides of the tweezer. This again indicates that the tweezer possibly requires additional stabilization by H-bonds or Van der Waals interactions to neighbor molecules in the crystal for stable binding, which is the prerequisite for sufficiently high occupancy and, therefore, visible electron density in a crystal structure. The methionine thioether and proline methylene groups, which point towards the tweezer cavity, are both regions of low polarity and high polarizability, ideal for additional dispersive interactions.

Tweezer binding is apparently also dependent on the pH, as P5CDH crystals grown at pH between 7.5 and 8.0 show a stronger electron density for CLR01 compared to a crystal grown at pH of 5.5 (author’s observation). This is most likely due to the increased protonation at low pH, resulting in a reduced negative charge of the phosphate group of the tweezer and, thus, less affinity to the positively charged region around the Met17 and neighboring lysine side chains (Figure 5C).

As mentioned previously, the region around Lys464 of P5CDH was found to resemble the binding epitope of CLR01 around Lys214 in the 14-3-3 complex (PDB ID 5OEH). Contrary to our expectation, in the P5CDH-CLR01 complex, no electron density for a tweezer molecule could be found next to Lys464. In all four monomers, the side chain of Lys464 is pointing into solvent channels (Appendix A), and in at least monomers B, C, and D, there would be sufficient space for a tweezer molecule between Lys464 and the next protein neighbor. The most probable reason for the failure of CLR01 binding at Lys464 is the configuration of the residues surrounding this lysine. The side chains of Tyr213 and Glu492 are probably too close to Lys464, so they sterically interfere with the binding of CLR01. Theoretically, the tweezer could tilt slightly to avoid the putative clash with Glu492, and the Tyr213 side chain could adopt a different rotamer to make space for the CLR01 molecule, but apparently, the binding energy of CLR01 is not sufficient to trigger this change in conformation, corroborating the relatively low affinity of the unmodified tweezer molecule. This result does not exclude weak or transient binding of the tweezer to this lysine side chain but shows that under the given conditions, the tweezer does not bind to Lys464 of P5CDH in a defined orientation with sufficiently high occupancy to be detectable in the crystal.

### 2.4. Structure of P5CDH with the Monophosphate Tweezer

Co-crystallizing P5CDH with the monophosphate tweezer (MPT) instead of the diphosphate tweezer CLR01 resulted in a different crystal form that diffracted to 1.2 Å (Table 2), with the physiological dimer (instead of a tetramer) in the asymmetric unit (Figure 6). Very clear electron density for a single MPT molecule was found next to Lys402 of monomer A, with the single phosphate group at the ε-amine of the side chain of Lys402 pointing into solvent and the hydroxyl group facing the α-helix (Figure 6, Appendix A). In addition, H-bonds between the phosphate group of the tweezer to the side chain of Lys52 of the neighbor protein molecule facilitate the formation of a noncovalent supramolecular bridge between the protein containing the hosted lysine side chain and the neighbor molecule. Lys402 is located opposite the N-terminus and dimerization hairpin of P5CDH, which forms the binding site for the CLR01 discussed above.

Interestingly, no electron density corresponding to the monophosphate tweezer can be observed next to Lys402 of monomer B. As shown in Figure 6, the environment in the crystal around Lys402 of monomer B is different compared to monomer A and does neither provide space for the tweezer to bind to Lys402 nor a H-bond acceptor for the tweezer’s phosphate group to stabilize binding. The closest contacts to a symmetry-related molecule are Pro120, Ala122, and Asp123. A superimposition of monomer A to monomer B shows the putative clash of the MPT molecule with Pro120 of the symmetry-related molecule and the lack of positive charge to stabilize the tweezer at this position (Figure 6D,E). This asymmetric binding of the tweezer to the two practically identical P5CDH molecules stresses the importance of crystal packing that very often competes with (especially sterically demanding) supramolecular ligands and either prevents their binding to the protein inside the crystals, or even crystallization itself if the affinity is sufficiently high to overcome the crystal packing forces.

It would be interesting to know if the presence of MPT directed the crystallization towards a monoclinic space group, as opposed to the hexagonal space groups of apo P5CDH and P5CDH-CLR01. This cannot currently be decided since the crystallization conditions of P5CDH-MPT lack the ammonium salt component (Table 1), so the underlying cause may be both the ligand and/or the different salt conditions or either one of them (Appendix A).

Since at least one MPT was bound at Lys402 of P5CDH, and since the solvent channels seem to be sufficiently large, this raises the question of why CLR01 did not bind at this position. A superimposition of CLR01 on the P5CDH-MPT crystal structure shows that the second phosphate group would clash with the protein surface (Appendix A), illustrating the more demanding sterical requirements for CLR01 binding. This is an important lesson from the systematic crystallization study with mono- vs. diphosphate tweezer: The second phosphate group causes a severe steric and possibly also electrostatic repulsion when the tweezers thread onto a lysine or arginine residue and closely approach the protein surface. This steric clash can be avoided by truncating one phosphate, which is not strictly necessary for the recognition of the positive charge of the amino acid side chain. It may be added that the deletion of the second phosphate severely lowers the solubility of the resulting tweezer derivative in an aqueous buffer.

Similar to the P5CDH-CLR01 structure, no electron density corresponding to the tweezer molecule is detected at the predicted 14-3-3-like binding epitope at Lys464. Due to the different space group of the MPT co-crystal structure of P5CDH, the side chain of Lys464 is located at a narrow solvent channel (Appendix A), indicating that either the competition by crystal packing forces is sufficiently strong to displace the MPT, or that the affinity of the MPT to this epitope is too weak to begin with (i.e., the MPT would then not even bind in solution). It may be speculated that, again, the orientation of neighboring side chains is unfavorable, as is the case for the CLR01 molecule, although the MPT is sterically much less demanding than the CLR01 due to the missing phosphate group.

## 3. Discussion

Taken together, the tweezer proved to be quite versatile in its choice of binding sites on protein surfaces. Although it clearly prefers well accessible lysine and arginine residues, the tweezer seems not restricted to these, but may also accommodate other guests like methionine, or even occupy appropriately shaped free space between protein domains, whenever additional stabilization by hydrogen bonds or dispersive interactions is possible. Such stabilization has been discussed as a means to facilitate crystallization and even trigger self-assembly of proteins [14].

The first crystal structures solved in complex with a tweezer molecule were 14-3-3 proteins with an advantageous shape and crystal packing, where relatively large solvent channels allowed the accommodation of bulky ligands. To extend the range of target proteins, a systematic search of potential binding sites for the diphosphate tweezer CLR01 was conducted in the protein database. A promising candidate was the enzyme P5CDH (Δ^1^-pyrroline-5-carboxylate dehydrogenase), where the surroundings of Lys464 feature a similar backbone conformation of the α-helix containing the lysine residue, as seen in a published 14-3-3-CLR01 structure (PDB ID 5OEH, Figure 7A–E). Surprisingly, the obtained crystal structures of P5CDH with CLR01 or even with the sterically less demanding monophosphate tweezer (MPT) did not show the tweezers bound to the expected Lys464, but instead the supramolecular ligands were either found wrapped around a methionine residue that belongs to a cloning extension of one of the N-termini of P5CDH (CLR01) or bound to another lysine (Lys402, MPT).

In more detail, the CLR01 molecule in P5CDH is surrounded by the N-terminus containing Met17 and the dimerization β-hairpin and forms H-bonds to a neighboring molecule of the asymmetric unit (Figure 5). This proves that CLR01 can actually host side chains that are different from lysine. In addition, the binding of CLR01 can rigidify previously unstructured regions, as seen in the P5CDH-CLR01 structure, where the N-terminus shows clear density at the tweezer-bound Met17, whereas, in the other three molecules in the asymmetric unit where no tweezer is bound, the N-termini are still disordered. This situation is similar to the 14-3-3-CDC25C peptide-CLR01 structure (PDB code 5M36, 5M37), where an arginine of the peptide is clamped by the tweezer and thus rigidified (Figure 3). In contrast, the monophosphate tweezer is actually bound at a lysine residue (Lys402) of P5CDH, but it is also stabilized by a neighboring monomer in the crystal (Figure 6A–C). This P5CDH-MPT complex crystallized in a previously unknown crystal form with a different environment at Lys402 as compared to the CLR01-P5CDH complex structure (Appendix A). In both crystals, only one tweezer molecule is seen in the asymmetric unit despite the presence of two and four monomers of P5CDH, respectively, revealing an asymmetric binding mode of the tweezer to P5CDH, depending on the non-equivalent environment of the tweezer binding sites in the monomers.

Three example structures of lysine side chains in complex with a tweezer are now available: two structures of 14-3-3 with CLR01 (5OEH and 5M37), and one of P5CDH with the monophosphate tweezer (P5CDH-MPT). The secondary structure constraints are quite different for the two tweezer variants: CLR01 with the two bulky phosphate groups is found at the end or kink of an alpha-helix, indicating that it most likely cannot bind to a lysine residue located within an uninterrupted helix since one phosphate group would always clash with the helix backbone. In contrast, the MPT in the P5CDH complex structure is actually found at a lysine side chain within an alpha-helix, where the smaller hydroxyl group that replaces one of the phosphates can readily be accommodated (Figure 7H,I,K,L). The monophosphate tweezer is, therefore, less demanding on the exact three-dimensional shape of the binding sites and might potentially have more (and obviously different) binding sites than CLR01. Interestingly, the affinities of mono- versus diphosphate tweezers to suitable ligands (e.g., lysine and arginine) appear to be similar [25]. In this respect, the truncated monophosphate tweezer may prove advantageous when close contact with lysines on sterically demanding secondary structure elements is desired. The MPT might thus prove to be a more versatile lysine/arginine-binding ligand.

The electrostatic surface representations of the tweezer complexes (Figure 7) highlight the generally demanding space requirements of the supramolecular tweezers, as well as their preference for positively charged regions that attract the strongly negative phosphate groups.

A more detailed analysis of the tweezer binding modes using the CCP4 PISA software [32] revealed that the largest hydrophobic Van der Waals interactions between the CLR01 and P5CDH monomer A are from the Met17 side chain bound inside the aromatic ring system, which is additionally clamped between two lysine residues that provide the positive charges to the region around the methionine (Figure 5C–F). Pro189 and Pro190 of the P5CDH dimerization β-hairpin sit on top of the tweezer, also providing additional hydrophobic interaction areas. The interface of CLR01 to monomer A buries 487 Å^2^ of the solvent-excluded surface and covers about 69% of the tweezer’s surface. The side chain of Ala33 of monomer D is located between both phosphate groups at the central ring of the tweezer, also contributing to the hydrophobic surface. The total area buried by the symmetry neighbor is 191 Å^2^, corresponding to approx. 27% of the tweezer surface, which is a very significant contribution to the stabilization of the CLR01 position in the crystal.

The surface around P5CDH-Lys402 bound by the monophosphate tweezer (Figure 6, Appendix A) is also strongly positively charged by the ε-amine of the clamped lysine and the neighboring arginines (Arg399 and the symmetry-related Lys5), and shows a surrounding “moat” that accommodates the ring system. Hydrophobic contributions to the buried surface in the P5CDH complex originate from the aliphatic part of the clamped lysine residue Lys402 and the side chains of Trp403 and His406. In total, the monophosphate tweezer buries an interface area of 371 Å^2^ and 140 Å^2^ to monomer A and the symmetry neighbor, respectively, i.e., slightly less area compared to the CLR01 at Met17.

In the 14-3-3-CLR01 structure with CLR01 at Lys214 (PDB ID 5OEH), there is a cleft in the 14-3-3 surface that accommodates the second phosphate (Figure 3C and Figure 7). Interestingly, this region formed by Glu210, Asp211, and Asp215 is negatively charged, indicating that the favorable interactions due to the shape complementarity and presence of the positive Lys214 and Arg224 are sufficient to overcome this putatively unfavorable interaction. These side chains interact with the phosphate via coordinated water molecules. In the 14-3-3 complex, Van der Waals interactions are observed to Leu218 and Tyr213 of monomer A, as well as to the side chains of Phe198, Met202, Thr217, and Met 220 of the symmetry neighbor, amounting to a total buried interface area of 289 Å^2^ and 173 Å^2^ to monomer A and the symmetry neighbor, respectively. This corroborates the apparent requirement of the tweezer molecule to be stabilized by the surrounding protein molecules in the crystal.

The second example of CLR01 bound to a lysine (PDB ID 5M37) allows the comparison of this binding mode with the previously described 14-3-3-CLR01 structure (5OEH). The CLR01 molecule is encircling Lys74 that is located at the end of an alpha-helix again, with one of the phosphates pointing into the solvent, and the other one is inserted into a quite negatively charged cleft formed by the side chains of Gln77, Glu73, and the symmetry-related Gln161 (Figure 3B,C). In contrast to 5OEH, the protein-bound phosphate group does not “cap” the end of the helix but sits on the side of the helix backbone. This might give the impression that the CLR01 could bind to the middle of a helix in this position, but if the helix were continued, the aromatic ring system would clash with the helix backbone. The outside of the ring system contacts the side chains of Met78 and Glu5, as well as Gln8 of monomer A, and on the other side, the backbone of the symmetry-related Glu159. The tweezer at Lys74 buries 317.5 Å^2^ with the monomer B that supplies the Lys74 and 109.9 Å^2^ with the symmetry-related monomer B. The contact to monomer A via Gln8 is very minor, with only 24 Å^2^. Thus, in this case, the CLR01 molecules are predominantly bound to the protein that provides the lysine, although the position of the ring system is very likely affected by the close vicinity of the neighboring protein molecule.

In summary, in all available structures with bound tweezer molecules (P5CDH, 14-3-3 σ and ζ), the “small” artificial ligand is stabilized by neighboring molecules in the crystal lattice. Quite likely, the interaction of the ligand with a neighboring protein molecule is needed to provide additional stabilization to the tweezer molecule to be visible in the electron density, as also very frequently observed for other low molecular weight compounds in the protein data bank [33]. Importantly, this does not contradict the transient, low-affinity binding of the tweezers to other surface-exposed lysine residues, as observed in solution. This highlights the need to increase the affinity and specificity of the tweezers by, e.g., attaching peptide moieties as described above.

A major problem of co-crystallizing tweezer compounds in general is the competition of the protein side chains with positive ions in the crystallization buffer that would fit into the aromatic ring system of the CLR01 molecule. The pore formed by the Van der Waals surfaces of the tweezer ring has a diameter of approximately 4.4 Å, so that calcium or sodium ions with Van der Waals radii of approximately 2.3 Å could fit in the center of the cavity. A magnesium ion is slightly smaller than calcium and sodium (1.7 Å) and similar to the size of the ε-amino group of a lysine (approx. 1.8 Å). The preferred position of a positive ion next to a symmetrical diphosphate tweezer molecule would, of course, be asymmetric and closer to either one or to the other phosphate moiety, but it would be located sufficiently close to the ring center to compete with, e.g., a lysine side chain, as judged by the observed electron densities. Furthermore, the lysine side chain will, of course, be coordinated by water molecules, further diminishing its interaction strength with the tweezer. On the other hand, the center of the cavity of the CLR01 is quite hydrophobic, so the accommodation of the hydrophobic “stem” of the lysine with a positive charge at the end is probably preferred at equal concentrations of the competitors. The crystallization conditions of the 14-3-3-CLR01 complex (5OEH) contain 1.26 M tri-sodium citrate, meaning that sodium most likely is a weaker competitor, probably due to its single charge, similar to the ammonium ion, which allows tweezer binding to lysine or methionine at 170 mM ammonium acetate (5M36, 5M37) or 200 mM ammonium sulfate (P5CDH) (Table 1). However, at sufficiently high concentrations, especially highly charged cations like calcium or magnesium will most likely show a competitive effect with lysine or arginine for the tweezer molecule. This consideration is also important for in vivo experiments, where tweezer molecules might encounter intracellular concentrations of up to 25 mM magnesium [34,35,36]. However, the calcium concentration in cells is supposedly much lower at 50–200 nM, so it is expected not to interfere with the binding of the tweezer [37].

## 4. Conclusions

Supramolecular tweezers can accommodate various host molecules in their central cavity, ranging from ions over lysine and arginine side chains to hydrophobic methionine side chains. Their versatile binding usually exhibits relatively low affinities in the low micromolar range, resulting in the need for stabilization by neighboring molecules to make them visible in a crystal structure. Increasing their specificity and affinity by attaching, e.g., peptides to the aromatic ring system, leads to a much wider range of addressable biological targets, as shown in the crystal structures of 14-3-3-proteins with such hybrid molecules. So far, there is no evidence that the (modified) tweezer molecules could successfully compete with the crystal packing forces, although it was noted that a high concentration of CLR01 actually can prevent crystallization by obstructing lysines that would have been in crystal contacts on the protein surface (author’s observation). Thus, in general, the space in the solvent channels of the co-crystals must be sufficiently large to allow the binding of the bulky tweezer molecules, which limits the applicability of crystallography for those molecules.

Especially in the field of 14-3-3 adaptor proteins, the biological relevance of tweezer binding to hotspots on protein surfaces has been demonstrated through the use of X-ray analysis: here, protein–protein interactions could be inhibited by a sterical clash with cargo proteins (14-3-3σ); on the other hand, the stabilization of PPIs was observed by tweezer docking to hydrophobic surface residues with concomitant lysine or arginine inclusion (14-3-3ζ); finally, existing natural peptide signals were strengthened by simultaneous tweezer docking (14-3-3σ). Generally, the truncated version of a monophosphate tweezer seems to be advantageous for improved threading onto side chains of globular proteins. Crystal structures thus revealed the exact binding geometry of tweezers on proteins and confirmed the postulated inclusion binding mode with additional ion pairing (14-3-3σ, 14-3-3ζ). They also showed, in great detail, how PPIs can be blocked and strengthened by single tweezers and proved the feasibility of rational design of conjugated tweezer-peptide hybrids with superior affinities and selectivities. Finally, X-ray crystallography suggested further improvement by truncating one phosphate moiety for enhanced lysine and arginine inclusion on the protein surface. In ongoing collaborations between structural biologists and supramolecular chemists, these lessons will be applied in the advanced tweezer design and hopefully lead to potent protein surface binders with predictable binding properties and increased biological as well as medicinal impact.

## 5. Methods

P5CDH was expressed and purified as described in Srivastava et al., 2012 [31]. The N-terminal His-tag of a pET28 plasmid with a cleavable thrombin site leads to the N-terminal sequence MGSSHHHHHHSSGLVPRGSHMTGAGL…, i.e., the P5CDH sequence starts at Thr17, the preceding amino acids are a cloning artifact. All CLR01 crystals were grown from PEG with either ammonium sulfate, ammonium acetate, or ammonium citrate pH 7.5–8 in space group *P*6_5_ with a dimer-of-dimers in the asymmetric unit. The crystal used for structure determination was grown in Nextal (Qiagen, Hilden, Germany) screen Classics II H4 0.2 M ammonium citrate, 20% PEG 3350 at 20 °C. The crystal was flash-frozen in liquid nitrogen, and the diffraction data were collected at 100K on a Pilatus 6M detector to 2.6 Å resolution at the SLS synchrotron in Villigen, Switzerland, in space group *P*6_5_ with a unit cell of a = 149.14 Å, b = 149.14 Å, c = 191.27 Å. Four monomers per asymmetric unit (AU) form a dimer-of-dimers.

For P5CDH with the monophosphate tweezer (MPT), box-shaped crystals were grown under Classics II screen conditions D8 (0.1 M HEPES pH 7.5, 25% PEG 3350) or D9 (0.1 M Tris pH 8.5, 25% PEG 3350) after 14 to 21 days at 20 °C. Crystals of both conditions belonged to space group *P*2_1_, with unit cell parameters of a = 71.7 Å, b = 84.7 Å, c = 85.5 Å, and γ = 103.3°. This is a new crystal form in space group *P*2_1_ as the unit cell parameters differ from the published structures with PDB ID 4OE5 (*P*2_1_ with unit cell a = 92.0 Å, b = 121.3 Å, c = 93.4 Å, and γ = 104.2°, where the crystal packing would not be compatible with tweezer binding (Appendix A) as well as PDB IDs 3V9K/3V9J/3V9L with an orthorhombic space group and unit cells of approximately a = 85 Å, b = 94 Å, c = 132 Å). On a side note, a PEG molecule formed a ring-like structure somewhat resembling a tweezer in the apo P5CDH structure 4OE5 around Lys93 of all four monomers in the asymmetric unit.

The crystal from condition D9 was flash-frozen in liquid nitrogen, and the data were collected at 100 K on a Pilatus 6M detector to 1.2 Å resolution at the SLS synchrotron in Villigen, Switzerland.

The data sets were integrated using XDS [38] and scaled with XSCALE [38]. The structures were solved via molecular replacement with PHASER (CCP4 suite) [39]. Refinement with REFMAC [39] and PHENIX [40] and model building with COOT [41] resulted in models with good geometries. All figures were prepared using either COOT or PYMOL [42].

Interestingly, an additional electron density is observed in the active site of monoclinic P5CDH next to the catalytic residue Cys348 in both monomers. This position is also known to bind the P5CDH inhibitor L-glutamate, where the carboxylate mimics the aldehyde group of the natural substrate γ-glutamate semialdehyde [43]. A planar ring shape with a forked extension that is positioned at a hydrogen bonding distance next to hydrophilic groups would be consistent with benzoic acid. This molecule is often found in crystal structures and could either be co-purified from E.coli or, probably more likely, could be leaked out of the plastic crystallization plates or other plastic products used during purification (plastic softener degradation product).

## Figures and Tables

**Figure 1 molecules-29-01764-f001:**
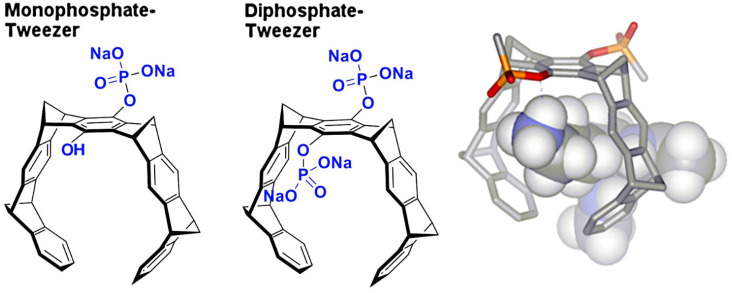
The figure shows tweezer subtypes: monophosphate tweezer (MPT, (**left**)) and diphosphate tweezer (CLR01, (**center**)). The electron-rich hydrophobic cavity can accommodate elongated side chains of amino acids like lysine and arginine, with the phosphate group forming an ion pair as well as hydrogen bond to the positively charged ε-ammonium group of lysine (**right**) or the δ-guanidinium head of arginine.

**Figure 2 molecules-29-01764-f002:**
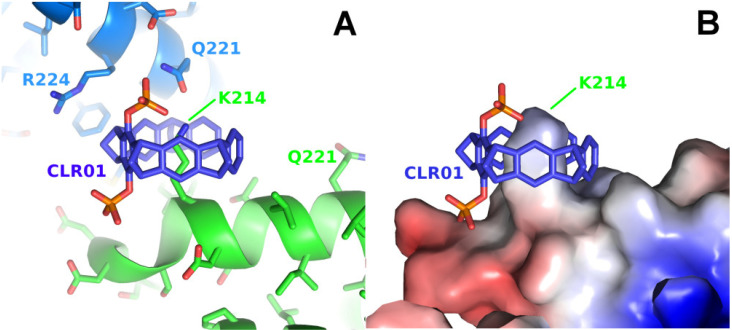
The figure shows the crystal structure of a 14-3-3 protein with CLR01. (**A**): Detail of the CLR01 (purple-blue) binding site next to Lys214 of 14-3-3 sigma (5OEH, green), with the symmetry-related molecule colored in light blue, where Gln221 and Arg224 are contributing to the interface with the tweezer. (**B**): The surface representation shows the shape complementarity of the CLR01 to the surface, with the bulky phosphate group accommodated by the kink formed at the end of the alpha helix 213–230 in 14-3-3. The surface is colored by the electrostatic surface potential (APBS plugin in Pymol; red/blue represents ±5 k_B_T/e), revealing that apparently, the shape complementarity for the phosphate next to the negatively charged surface is more important than the charge complementarity. In contrast, the second phosphate is bound canonically to the positively charged Lys214 side chain.

**Figure 3 molecules-29-01764-f003:**
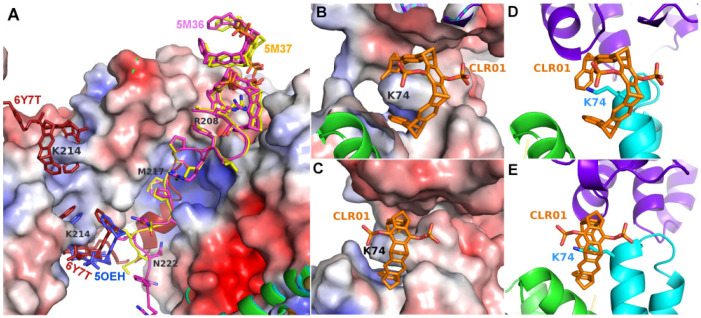
The figure shows the crystal structures of 14-3-3 proteins with CLR01. (**A**): Superimposition of 14-3-3-CLR01 complex (5OEH, blue), 14-3-3-tweezer-CDC25C complex (5M36, magenta, and 5M37, yellow), and 14-3-3-tweezer-ExoS-hybrid (6Y7T, firebrick). (**B**,**C**): A fourth tweezer molecule in 5M37 is bound to Lys74 and—similar to the CLR01 at Lys214 in 5OEH—also at the end of an alpha helix, with very good shape complementarity. (**D**,**E**): CLR01 bound to Lys74 at the helix terminus (cyan) packs against a symmetry-related molecule shown in purple. The orientation of CLR01 relative to the helix is different from the position next to Lys214 in 5OEH, which highlights the influence of the crystal packing on the specific orientation of the tweezer molecule. Electrostatic surface potentials were calculated by the APBS plugin in Pymol; red/blue represents ±5 k_B_T/e.

**Figure 4 molecules-29-01764-f004:**
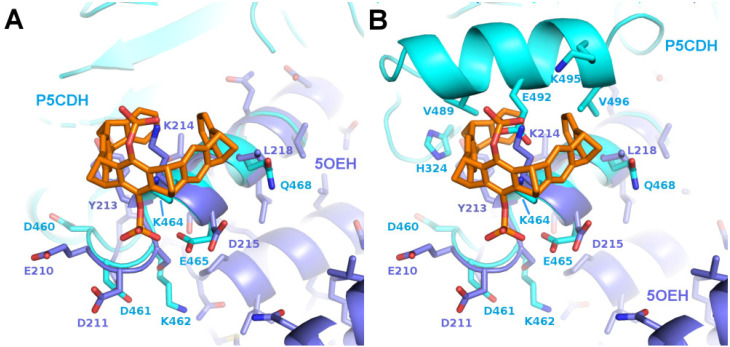
The figure shows a comparison of the predicted binding motif of P5CDH with the binding motif of 14-3-3. Superimposition of P5CDH (cyan) around Lys464 with 14-3-3 around Lys214 with bound CLR01 (5OEH, purple). (**A**): Corresponding segments are superimposed and parts of P5CDH omitted for clarity, (**B**): The complete P5CDH is shown, including the side chains next to the tweezer molecule. The putative clashes with the P5CDH-helix 487–496 could theoretically be relieved by a slight rotation of the CLR01 molecule.

**Figure 5 molecules-29-01764-f005:**
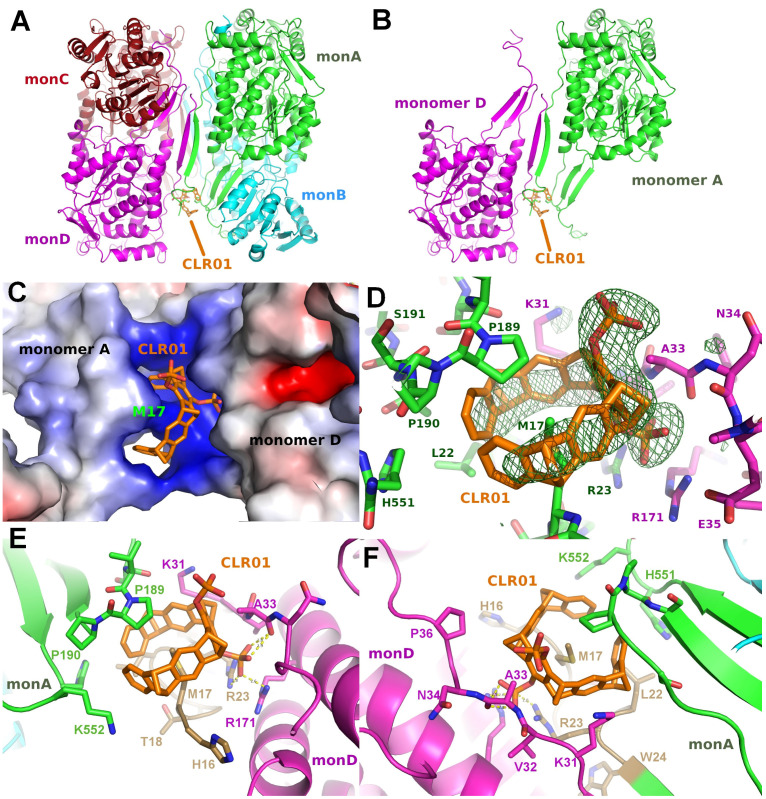
The figure shows the crystal structure of P5CDH with CLR01. (**A**,**B**): Overview of the asymmetric unit containing a dimer-of-dimers with CLR01 bound between the N-terminus and the dimerization hairpin of Monomer A (green) and Monomer D (magenta). Molecule B (cyan) of the physiologically relevant dimer does not form any contact with CLR01. Binding of the tweezer causes a slight relative rotation of the two dimers by 3.6° relative to the two physiological dimers of the apo form (PDB ID 3V9G). (**C**): Electrostatic surface representation shows the positively charged surrounding of the binding site of the tweezer molecule, which is wrapped around the side chain of Met17 with one of the phosphate groups buried in the cleft between monomers A and D. (**D**): Difference electron density (green, contoured at 3 sigma) for CLR01 bound to the Met17 side chain at the N-terminus of Monomer A. The tweezer is flanked by Lys31 of Monomer D (magenta). (**E**,**F**): Detailed view on the CLR01 binding around the side chain of Met17, clamped between the tip of the P5CDH dimerization β-hairpin formed by Pro189 and Pro190 (green sticks) and the N-terminus that contains the cloning artifact Met17 (gold). One phosphate group forms hydrogen bonds to the side chain of Arg171 and the backbone amide of Ala33 of monomer D (magenta). In the top view (**F**) of the aromatic ring system, the Lys31 and Lys552 side chains can be seen framing the tweezer molecule on both sides. Electrostatic surface potentials were calculated by the APBS plugin in Pymol; red/blue represents ±5 k_B_T/e.

**Figure 6 molecules-29-01764-f006:**
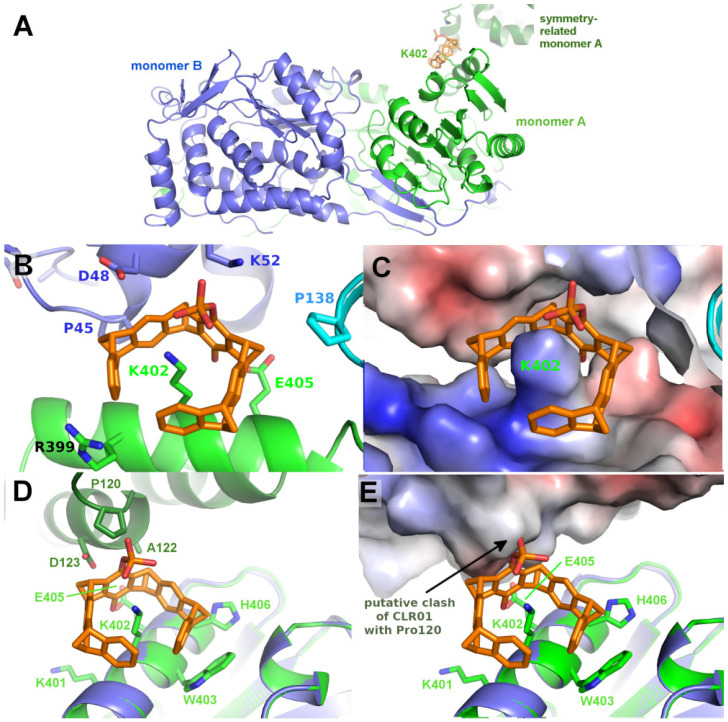
The figure shows the crystal structure of P5CDH with monophosphate tweezer. (**A**): Overview of the asymmetric unit of the P5CDH-monophosphate tweezer (MPT) complex consisting of the physiological dimer (shown in green/blue) and one molecule of MPT (orange sticks) bound to the side chain of Lys402. (**B**): Position of the MPT between monomer A (green) and the symmetry-related monomer B (blue). (**C**): same as (**B**), but with the molecular surface colored by the electrostatic surface potential, highlighting the strongly positively charged binding region around lysine 402. (**D**): Environment of the uncomplexed Lys402 in monomer B (blue) with a model of the monophosphate tweezer of monomer A (orange/green) superimposed. The modeled tweezer (orange sticks) shows that the crystal-symmetry-related molecule (dark green) is too close to allow binding at this position of monomer B, as the tweezer’s phosphate group would clash with Pro120 of the symmetry neighbor. (**E**): Electrostatic surface of the symmetry neighbor showing the putative clash of the tweezer’s phosphate with Pro120. Electrostatic surface potentials were calculated by the APBS plugin in Pymol; red/blue represents ±5 k_B_T/e.

**Figure 7 molecules-29-01764-f007:**
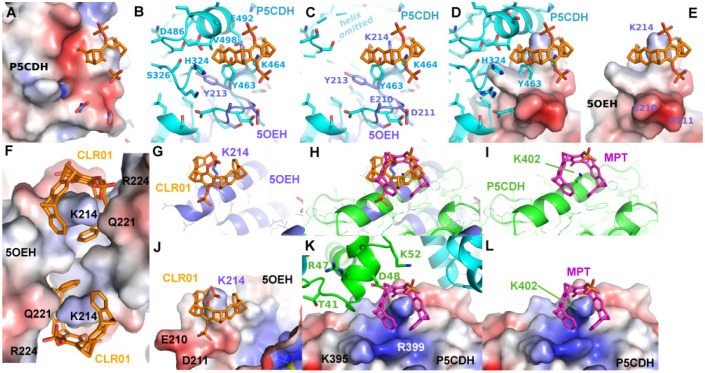
The figure shows a comparison of the observed and putative CLR01 binding modes in P5CDH and 14-3-3. (**A**–**E**): Comparison of the CLR01 binding epitope on Lys214 of 14-3-3 (purple) with the predicted epitope on P5CDH (cyan). CLR01 (orange sticks) is always shown in the position bound to Lys214 of 14-3-3. In (**C**), the P5CDH helix 487–496 is omitted for clarity. (See also Figure 4 for a different orientation of this superposition). A reorientation of CLR01 should be possible to avoid the putative clash with the P5CDH helix, and the Tyr463 side chain could adopt a rotamer that would allow binding of the tweezer, but apparently, the resulting affinity is too low to allow binding of the CLR01 molecule, although there should be sufficient space in the crystal packing. (**F**–**L**): Comparison of the monophosphate tweezer (MPT) binding mode to Lys402 in the middle of helix 394–408 of P5CDH (green) with the CLR01 binding mode to Lys214 of 14-3-3 at the end of helix 213–230 (purple). CLR01 is shown in orange sticks, and MPT in magenta sticks. F shows the remarkable shape complementarity of the surface of the two symmetry-related 14-3-3 molecules that surround the two CLR01 molecules, colored by electrostatic surface potential. Panels (**G**–**L**) show the superimposed helices of 14-3-3 and P5CHD in the same orientation, highlighting the different binding modes of the tweezers. The second phosphate of CLR01 (orange) forces the positioning at the end of the 14-3-3 helix (**J**), whereas the MPT (magenta) can bind at the lysine of an intact alpha helix, with the phosphate group pointing into the solvent. The strong positive charges of P5CHD around Lys402 (**K**,**L**) support the binding of the negative phosphate group. Panel H shows the two tweezers superimposed to illustrate the different positions relative to the alpha helix. Electrostatic surface potentials were calculated by the APBS plugin in Pymol; red/blue represents ±5 k_B_T/e.

**Table 1 molecules-29-01764-t001:** The table shows the crystallization conditions of the tweezer complexes.

Complex	Resol.	M	SG	Unit Cell	Guest	Crystallization Condition
P5CDH apo (3V9J/ 3V9K/3V9L)	1.3–1.5 Å	2	*P*2_1_2_1_2_1_	85, 94, 132 Å 90, 90, 90°	-	20–25% *w*/*v* PEG3350, 0.2 M LiSO_4_, 0.1 M Bis-Tris pH 6.5
P5CDH apo (3V9G)	2.5 Å	4	*P*6_5_	150.7, 150.7, 192.0 Å, 90, 90, 120°	-	22.5% PEG3350, 0.2 M (NH_4_)_3_ sulfate, 0.1 M HEPES pH 7.5
**P5CDH-CLR01**	2.6 Å	4	*P*6_5_	148.8, 148.78, 190.0 Å 90, 90, 120°	Met	0.2 M (NH_4_)_3_ citrate, 20% PEG 3350
**P5CDH-MPT**	1.2 Å	2	*P*2_1_	71.9, 85.1, 85.6 Å 90, 103.2, 90°	Lys	25%PEG3350, 0.1 M TRIS pH 8.5
P5CHD apo (4OE5)	1.95 Å	4	*P*2_1_	92, 121.3, 93.4 Å 90, 104.2, 90°	-	20–25% *w*/*v* PEG3350, 0.2 M MgCl_2_, 0.1 M HEPES pH 7–8
14-3-3-CLR01 (5OEH)	2.35 Å	1	*C*222	60, 157, 77 Å 90, 90, 90°	Lys	1.26 M tri-sodium citrate, 10%glycerol, 0.09 M Na-HEPES pH 7.5
14-3-3-CLR01+ non-coval. peptide (5M36/37)	2.35 Å	2	*P*2_1_2_1_2_1_	**5M36:** 71.5, 102.4, 112.8 Å, 90, 90, 90° **5M37:** 71.2, 88.2, 112.6 Å 90, 90, 90°	Arg, Lys	25.5% PEG 4000, 0.17 M (NH_4_)_3_ acetate, 0.085 M tri-sodium citrate pH 5.6, 15% glycerol (same condition for 5M35, 5M36 and 5M37)
14-3-3-CLR01-peptide-hybrid (6Y7T)	2.5 Å	4	*I*2	145.2, 63.1, 167.0 Å 90, 101.3,90°	Lys	20% PEG1000, 0.2 M Na acetate trihydrate, bis-tris-propane pH 7, 10% glycerol

The CLR01 and monophosphate tweezer (MPT) complexes of this work are highlighted in **bold** and are compared to crystallographic parameters and tweezer guests of published structures. M = molecules per asymmetric unit. P5CDH crystallized in the space group P6_5_ in the presence of ammonium salts. The crystals of the MPT complex were grown in 25% PEG3350 and Tris pH 8.5 or HEPES pH 7.5, lacking any additional salt components, and representing a new space group with four instead of two monomers per unit cell compared to the (also monoclinic) apo P5CDH structure 4OE5.

**Table 2 molecules-29-01764-t002:** The table shows data collection and refinement statistics.

	P5CDH-CLR01 (PDB 8RKQ)	P5CDH-MPT (PDB 8RKR)
Wavelength	0.97794	0.9188
Resolution range	48.21–2.6 (2.693–2.6)	42.13–1.2 (1.243–1.2)
Space group	*P*6_5_	*P*2_1_
Unit cell	148.8 148.8 190.0 90 90 120	71.9 85.2 85.6 90 103.1 90
Total reflections	654389 (62098)	2106606 (208643)
Unique reflections	72711 (7197)	312301 (31102)
Multiplicity	9.0 (8.6)	6.7 (6.7)
Completeness (%)	99.56 (99.08)	99.88 (99.85)
Mean I/sigma(I)	9.48 (0.77)	11.66 (0.76)
Wilson B-factor	73.60	14.31
R-merge	0.1317 (1.977)	0.08005 (2.148)
R-meas	0.1399 (2.103)	0.08677 (2.33)
R-pim	0.04687 (0.7122)	0.03313 (0.8898)
CC1/2	0.998 (0.656)	0.999 (0.344)
CC *	0.999 (0.89)	1 (0.716)
Reflections in refinement	72706 (7197)	312278 (31102)
Reflections for R-free	2097 (207)	3279 (326)
R-work	0.2311 (0.4023)	0.1725 (0.3326)
R-free	0.2713 (0.4090)	0.1914 (0.3463)
CC(work)	0.945 (0.640)	0.973 (0.645)
CC(free)	0.930 (0.514)	0.974 (0.612)
Non-hydrogen atoms	16936	9626
macromolecules	16829	8410
ligands	52	84
solvent	55	1132
Protein residues	2176	1064
RMS(bonds)	0.014	0.014
RMS(angles)	1.67	1.79
Ramachandran:		
favored (%)	94.51	97.92
allowed (%)	5.49	2.08
outliers (%)	0.00	0.00
Rotamer outliers (%)	3.08	0.66
Clashscore	3.25	3.79
Average B-factor	93.40	18.61
macromolecules	93.45	17.46
ligands	102.35	23.00
solvent	70.24	26.81

Statistics for the highest-resolution shell are shown in parentheses.

## Data Availability

All new data of this research have been submitted to the Protein Data Bank PDB and are accessible through the structure codes 8RKQ and 8RKR.

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
