# Peer review of "How Do Molecular Tweezers Bind to Proteins? Lessons from X-ray Crystallography"

_molecules, 2024, doi:10.3390/molecules29081764_

Round 1

Reviewer 1 Report

Comments and Suggestions for Authors

The authors of this manuscript present a detailed analysis of the binding mode of molecular tweezers, based on available crystal structures, as well as crystal structures that they determined and analyzed in the frame of the submitted work. These molecules are used to inhibit protein-protein interactions, or provide insight into the interaction site, and thus are of great interest for drug design. However, they are quite recalcitrant to crystallization in complex with their target protein, and thus the available structural information remains quite limited. Here, the authors analyze existing crystal structures of 14-3-3 proteins in complex with molecular tweezers and use another protein (P5CDH) as molecular probe to study the binding of molecular tweezes on its surface. Due to steric hindrances, the authors do not manage to co-crystallize P5CDH with a molecular tweezer at the anticipated location. Thus, one of the purposes of their work is not accomplished. They locate, however, two different sites, and analyze the two novel structures. The main findings is that a methionine can be accommodated inside a tweezer and that a monophosphate tweezer can be used when sterically demanding sites are under investigation

I think the manuscript can be accepted for publication after the following comments have been addressed.

Line 70: This is not a review paper, since the authors present novel experimental results.

Paragraph 107-116: again, here the authors should state that they have solved novel structures as well.

Lines 618-620: The authors shoud specify whether they crystallized the molecular tweezer alone and if yes provide data collection statistics.

Line 636: The crystal that grew in condition D9 was cryoprotected prior to freezing? Please state it if yes.

Data collection statistics for PDB code 8RKR: I would suggest reprocessing the data to CC1/2=0.5 at the outer shell, and check whether this improves the statistics.

The figures need to be improved in terms of resolution and also:

Figure 2A, 3: add labels for the amino acids discussed in the legend and the manuscript.

Figure 4: the side chain of K464 should be displayed. 4A can be omitted. The colors are appended incorrectly in the legend. The authors should comment on the E492 charge incompatibility with the phosphate group of CLR01. Y213 should be displayed as well.

Figure 5D: the amino acids should be labeled.

Line 279: explain cloning artifact.

Line 67: “the number of solved crystal structures”, the “s” should be removed in “crystals”.

The numbering of references should not be after the sentence end.

Reviewer 2 Report

Comments and Suggestions for Authors

Molecular tweezers can accommodate various residues in their central cavity. Here, Porfetye et al. investigated all known crystal structures with molecular tweezers. They found that the molecular tweezers can include lysine and arginine side chains, Moreover, they also solved crystal structures of two different tweezer variants in complex with the enzyme P5CDH, and found that the tweezers include a lysine and methionine side chain, respectively. Finally, they discussed the different binding modes and their implications for affinity and specificity, and the general problems in crystallizing protein complexes with artificial ligands.

Overall, this manuscript provides new experimental evidence and a summary of molecular tweezers, providing several structural and functional insights.

However, there are some major questions, for example:

1.     No label was labelled in several figures, such as should label Gln221, Arg224 and other important residues in Figure 2A. Please revise all the figures like Figure 7F. The phenomena are also found in Figures 2B, Figures 3B-3E, and Figures 5A-5D.

2.     Several figures look ugly, please remove unused lines and waters for clarity, such as sticks in the upper right of Figure 2A, unused lines of Figure 4, and all unnecessary waters of the Figures 6B and 6C。

3.     In this manuscript, electrostatic potentials are used in many places. The authors did not mention how the electrostatic surface calculations have been performed. Pymol vacuum electrostatic calculations should not be used to derive any scientific conclusions. Preferably, a freely available APBS software, the Adaptive Poisson-Boltzmann Solver, should be used for macromolecular electrostatics calculations. The label “Electrostatic” should be explained in the figure legend as “the surface potential”. The units for electrostatic potential are kBT/e.

4.     In Figure 5D, should label which sigma level of Difference electron density (green)? If it is low, the structure of CLR01 is not confident.

5.     In Table 2, the Mean I/sigma(I) values of the highest shell are too low, such as (0.77) and (0.76), So the resolution should be reduced.

Minor problems:

6.     In the line 36 of Page 1,  grammar error,  A prominent target are    should change to  A prominent target is  .

7.     In the line 42 of Page 1,  grammar error,  develop    should change to  developing .

8.     In the line 67 of Page 2,  grammar error,  crystals structures   should change to  crystal structures  .

9.     In the line 85 of Page 2,  spelling error,  accomodate    should change to  accommodate  .  Several similar errors were found in this manuscript.

10.  Many units should add a space after the values,  such as lines 579-580  25mM and 50-200nM.

11.  The character of the number of the space group in line 132 and Table 1 should be in italics.

12.  In the Abstract, the authors should add a sentence to show the importance of Molecular Tweezers.

13.  The position of all citing references should be before the period, please revise them all throughout the manuscript.

14.  The number 2 of MgCl2 should be subscript in Table 2 and Table 1.

Round 2

Reviewer 2 Report

Comments and Suggestions for Authors

 In this revision, the authors revised many errors and improved the manuscript greatly. One revision is needed.

 In Figure 5D, difference electron density contoured at 2 sigma is too low, usually at 3, at least 2.5.

Moreover, the figure looks ugly, please remove all unrelated density other than the difference electron density for CLR01.

Author Response

In Figure 5D, we now contoured the electron density at sigma level 3 and removed unrelated density. The new Figure 5 has been replaced in the manuscript, and the legend has been updated to read: ...contoured at 3 sigma...